# Sustained NPY signaling enables AgRP neurons to drive feeding

Yiming Chen[1,2†], Rachel A Essner[3†], Seher Kosar[3], Oliver H Miller[3], Yen-Chu Lin[3], Sheyda Mesgarzadeh[3], Zachary A Knight[1,2,3,4*]

[1]Kavli Institute for Fundamental Neuroscience, University of California, San Francisco, San Francisco, United States; [2]Neuroscience Graduate Program, University of California, San Francisco, San Francisco, United States; [3]Department of Physiology, University of California, San Francisco, San Francisco, United States; [4]Howard Hughes Medical Institute, University of California, San Francisco, San Francisco, United States

**Abstract** Artificial stimulation of Agouti-Related Peptide (AgRP) neurons promotes intense food consumption, yet paradoxically during natural behavior these cells are inhibited before feeding begins. Previously, to reconcile these observations, we showed that brief stimulation of AgRP neurons can generate hunger that persists for tens of minutes, but the mechanisms underlying this sustained hunger drive remain unknown (Chen et al., 2016). Here we show that Neuropeptide Y (NPY) is uniquely required for the long-lasting effects of AgRP neurons on feeding behavior. We blocked the ability of AgRP neurons to signal through AgRP, NPY, or GABA, and then stimulated these cells using a paradigm that mimics their natural regulation. Deletion of NPY, but not AgRP or GABA, abolished optically-stimulated feeding, and this was rescued by NPY re-expression selectively in AgRP neurons. These findings reveal a unique role for NPY in sustaining hunger in the interval between food discovery and consumption.
DOI: https://doi.org/10.7554/eLife.46348.001

**\*For correspondence:**
zachary.knight@ucsf.edu

[†]These authors contributed equally to this work

**Competing interests:** The authors declare that no competing interests exist.

## Introduction

AgRP neurons are a small population of cells in the arcuate nucleus of the hypothalamus that are critical for regulating food intake (*Andermann and Lowell, 2017*). These cells are progressively activated during food deprivation (*Betley et al., 2015*; *Chen et al., 2015*; *Hahn et al., 1998*; *Mandelblat-Cerf et al., 2015*), in part due to fluctuations in nutritionally-regulated hormones such as leptin and ghrelin (*Beutler et al., 2017*; *Blouet and Schwartz, 2010*; *Cowley et al., 2001*; *Cowley et al., 2003*; *Dietrich et al., 2013*; *Nakazato et al., 2001*; *Pinto et al., 2004*; *Schneeberger et al., 2013*). Optogenetic or chemogenetic stimulation of AgRP neurons promotes intense food seeking and consumption (*Aponte et al., 2011*; *Krashes et al., 2011*), whereas silencing of these cells leads to aphagia or starvation (*Gropp et al., 2005*; *Krashes et al., 2011*; *Luquet et al., 2005*). Thus AgRP neurons play a key role in the homeostatic negative feedback loop that links the need for energy to the stimulation of feeding behavior.

Although AgRP neurons are gradually activated during fasting, they are inhibited within seconds when a hungry animal sees or smells food (*Betley et al., 2015*; *Chen et al., 2015*; *Mandelblat-Cerf et al., 2015*). Paradoxically, this rapid inhibition of AgRP neurons often occurs before a single bite of food has been consumed and then persists for the duration of the ensuing meal. These observations raise the question of how AgRP neuron activity is able to drive food consumption, given that these neurons are inhibited during the act of feeding itself (*Chen and Knight, 2016*).

To reconcile these observations, we hypothesized that AgRP neurons may transmit a long-lasting signal that continues to drive food consumption even after AgRP neurons have been inhibited by

food cues (*Chen and Knight, 2016*). Consistent with this, we showed that brief optogenetic stimulation of AgRP neurons in the absence of food can drive voracious food intake tens of minutes later, long after AgRP neuron stimulation has been terminated (*Chen et al., 2016*). This indicates that AgRP neurons can generate an unusual 'sustained hunger drive' that outlasts their acute firing (*Figure 1A*). The mechanisms underlying this long-lasting potentiation of feeding behavior are unknown.

## Results

AgRP neurons release three neurotransmitters that can influence feeding: AgRP, NPY and GABA (*Andermann and Lowell, 2017*; *Hahn et al., 1998*; *Krashes et al., 2013*; *Tong et al., 2008*). We hypothesized that one of these neurotransmitters may play a specific role in the sustained hunger drive generated by AgRP neurons. To test this hypothesis, we blocked signaling through each of these pathways and then tested their role in AgRP neuron driven food intake (*Figure 1B*). To inactivate GABA signaling, we crossed AgRP-IRES-Cre mice to animals with a floxed allele of the GABA transporter *Slc32a1* (*Tong et al., 2008*) (referred to as GABA-); to block NPY signaling we used *NPY* knockout mice (*Erickson et al., 1996*) (NPY-); and for AgRP we used Agouti mice (AgRP-), in which the melanocortin pathway is inactivated by constitutive expression of a melanocortin 4-receptor antagonist (*Aponte et al., 2011*). We then measured the effect of optogenetic pre-stimulation (*Chen et al., 2016*) of AgRP neurons in each of these strains, or littermate controls.

We first stimulated AgRP neurons for 15 min in the absence of food, and then shut off the laser and allowed animals to eat for one hour. This pre-stimulation paradigm produced robust food intake in mice lacking GABA or AgRP signaling, and the magnitude of food consumption was similar between mutants and littermate controls (*Figure 1C*). In striking contrast, pre-stimulation of mice lacking NPY caused no increase in food intake (*Figure 1C*, bottom row). To measure the quantitative requirement for these neurotransmitters, we repeated this experiment with different durations of pre-stimulation (5, 15, 30 and 60 min; *Figure 1D*). We found that control mice, as well as mice lacking GABA or AgRP, showed a clear dose-response relationship between pre-stimulation duration and total food intake (*Figure 1E,F* and *Figure 1—figure supplement 1*). This increase in food intake was due to an increase in bout size, rather than bout number (*Figure 1—figure supplement 2*). In contrast NPY- mice showed no dose-dependent increase in food intake following longer durations of AgRP neuron pre-stimulation (*Figure 1G*). This behavioral defect was not due to a failure to activate AgRP neurons, because we observed robust Fos expression in AgRP neurons after optogenetic stimulation of NPY- mice (*Figure 1L*). Instead this indicates that NPY is specifically required for the sustained hunger produced by AgRP neuron activation.

To examine the kinetics of AgRP neuron driven hunger, we pre-stimulated AgRP neurons for five minutes and then inserted a delay of varying lengths between the termination of stimulation and the availability of food (15, 30, or 60 min; *Figure 1H*). We found that all genotypes except for NPY- mice exhibited robust food consumption that gradually declined with increasing delay duration up to 60 min (*Figure 1I–K*). By pooling data from all control animals, we estimated that this hunger signal has a rise time of approximately four minutes (logarithmic growth; N = 23; R-Square = 0.72; *Figure 1M*) and a decay-time of approximately 47 min (exponential decay; N = 19; R-Square = 0.87; *Figure 1N*). This indicates that AgRP neuron activity rapidly generates a durable, NPY-dependent signal that drives feeding behavior.

AgRP neurons promote not only food consumption but also behaviors that lead to food discovery (*Krashes et al., 2011*). We therefore examined the requirement for each of these neurotransmitters in AgRP driven appetitive behavior. To do this we used a progressive ratio three lever-pressing task, in which delivery of each subsequent pellet of food requires three additional lerel presses (*Figure 2A*). We observed vigorous lever pressing following pre-stimulation of AgRP neurons in AgRP- and GABA- mice, as well as controls (*Figure 2B–F*). In contrast pre-stimulation had no effect on operant responding in animals lacking NPY (*Figure 2B,C,F*). This indicates that NPY is critical for the ability of AgRP neurons to drive food seeking behavior as well as food consumption.

To investigate with greater temporal resolution the requirement for NPY in AgRP driven feeding behavior, we developed a paradigm that alternates between concurrent and pre-stimulation. In this

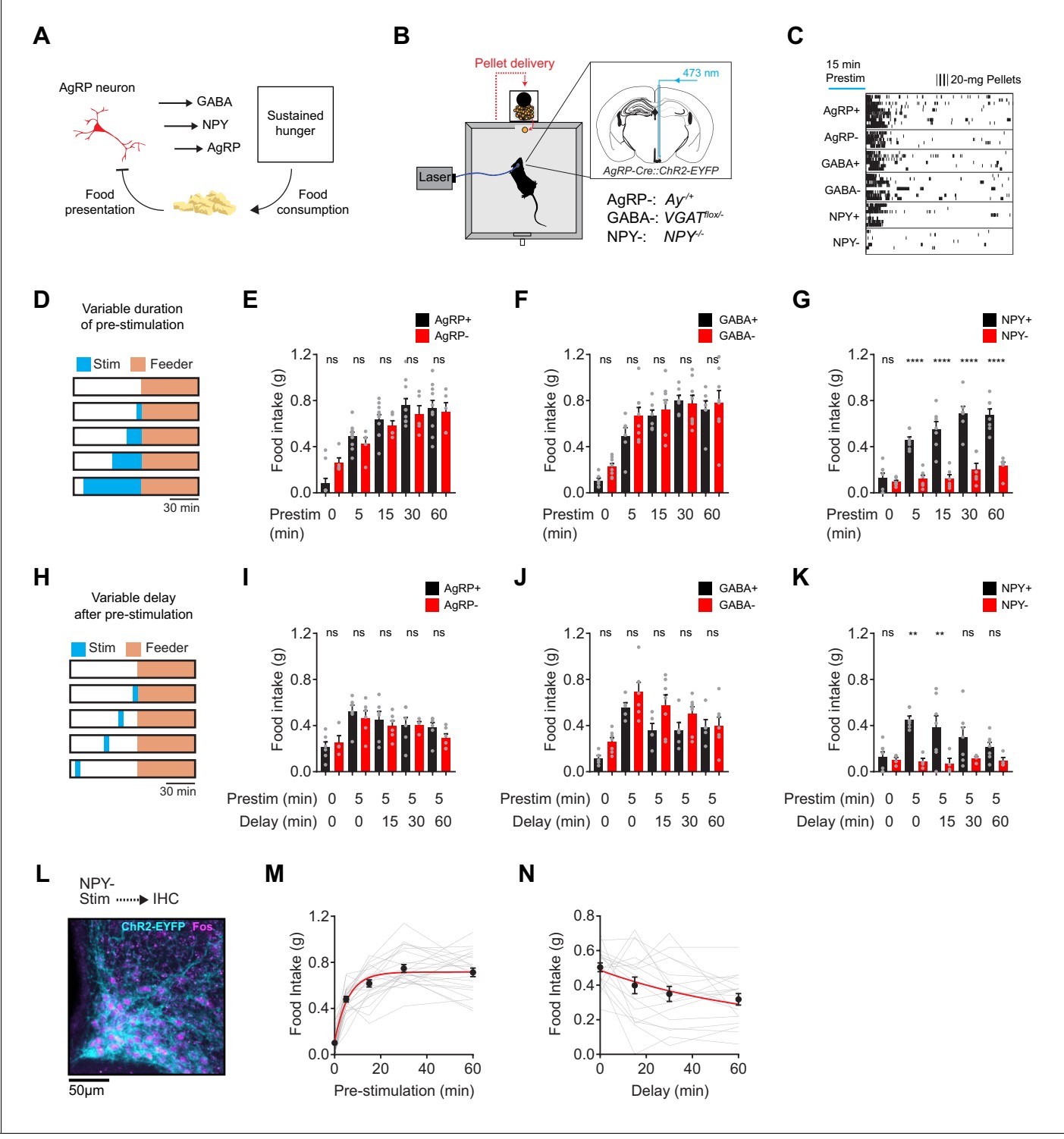

**Figure 1.** NPY is uniquely required for sustained hunger driven by AgRP neurons. (a) AgRP neurons are rapidly inhibited by food presentation but nevertheless promote food intake through a sustained hunger signal. (b) System for measuring food intake in response to activation of AgRP neurons in animals with blockade of signaling through AgRP, GABA or NPY (indicated as -) or littermate controls (indicated as +). (c) Raster plot showing pellet consumption of mice with different genotypes in response to pre-stimulation of AgRP neurons (15 min). Each row shows a representative trial from an individual mouse. (d) Feeding behavior paradigm with variable pre-stimulation lengths (blue). (e-g) Food intake (1 hr) following pre-stimulation of variable lengths. Genotypes: AgRP+ (black; N = 10) and AgRP- (red; N = 5) (e), GABA+ (black; N = 6) and GABA- (red; N = 8) (f), and NPY+ (black; N = 7) and NPY- (red; N = 6) (g). (h) Feeding behavior paradigm with variable delay length inserted between the end of pre-stimulation and start of

*Figure 1 continued on next page*

*Figure 1 continued*

food availability. (i-k) Food intake (1 hr) following pre-stimulation (5 min) and delay of varying lengths. Genotypes: AgRP+ (black; N = 6) and AgRP- (red; N = 6) (i), GABA+ (black; N = 5) and GABA- (red; N = 7) (j), and NPY+ (black; N = 7) and NPY- (red; N = 4) (k). (l) Image of ARC from NPY- mice that received laser stimulation (30 min) in the absence of food before perfusion. Sample was stained for ChR2-EYFP (cyan) and Fos (magenta). (m) Logarithmic growth model of relationship between food intake and pre-stimulation duration (N = 23 mice). (n) Exponential decay model of the relationship between food intake and delay length (N = 18 mice). Error bars in e-g, i-k, m and n represent mean ± SEM. Holm-Sidak multiple comparisons test was used to report adjusted P-values in e-g and i-k. *p<0.05, **p<0.01, ***p<0.001, ****p<0.0001 or ns (not significant) compares adjacent control (black) and experimental (red) group with same pre-stimulation protocol.

DOI: https://doi.org/10.7554/eLife.46348.002

The following figure supplements are available for figure 1:

**Figure supplement 1.** Temporal pattern of pellet consumption in response to different pre-stimulation protocols.
DOI: https://doi.org/10.7554/eLife.46348.003
**Figure supplement 2.** Bout analysis of food intake in response to pre-stimulation a-c.
DOI: https://doi.org/10.7554/eLife.46348.004

paradigm, animals are given constant access to food, and then subjected to two minutes of stimulation (20 Hz; Laser ON) followed by two minutes of no stimulation (Laser OFF) in a pattern that repeats for two hours (*Figure 3A*). In control mice, this protocol stimulated robust food consumption (*Figure 3B,C*). Importantly, food intake was equally distributed between the Laser ON and OFF epochs (*Figure 3C–I*), indicating that wild-type animals do not distinguish the boundary between these two-minute intervals, presumably due to the sustained hunger produced by AgRP neuron stimulation.

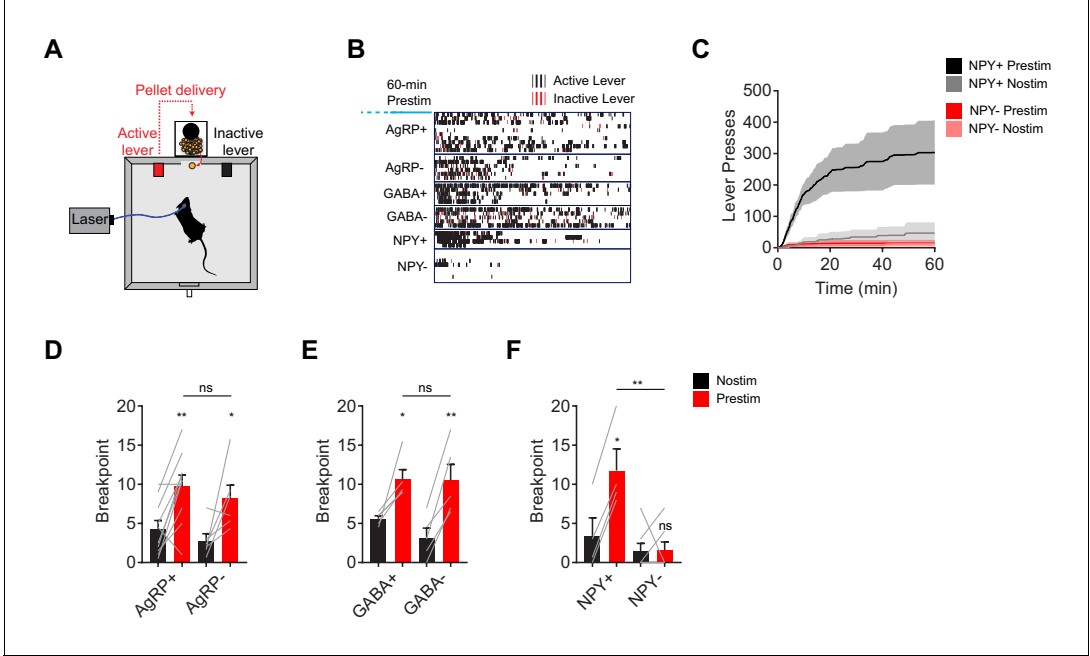

**Figure 2.** NPY, but not GABA or AgRP, is required for the sustained effect of AgRP neuron stimulation on appetitive behavior. (a) Schematic for progressive ratio three tasks, in which each successive pellet of food requires the mouse to press the lever three more times than the previous pellet. (b) Raster plot showing active (black) and inactive (red) lever presses of mice of different genotypes following pre-stimulation of AgRP neurons (1 hr). (c) Cumulative number of lever presses for NPY+ (N = 4) and NPY- (N = 7) mice in response to pre-stimulation or no stimulation of AgRP neurons. d-f, Breakpoint of AgRP+ (N = 10) and AgRP- (N = 6) (d), GABA+ (N = 5) and GABA- (N = 5) (e), and NPY+ (N = 4) and NPY- (N = 7) (f) mice in progressive ratio three after pre-stimulation (red) or no stimulation (black). Filled area in c and error bars in d-f represents mean ± SEM. Holm-Sidak multiple comparisons test was used to report adjusted P-value in d-f. *p<0.05, **p<0.01, or ns (not significant) compares adjacent control (black) and experimental (red) group with same pre-stimulation protocol, or compares pairs of groups that are indicated by horizontal lines above.
DOI: https://doi.org/10.7554/eLife.46348.005

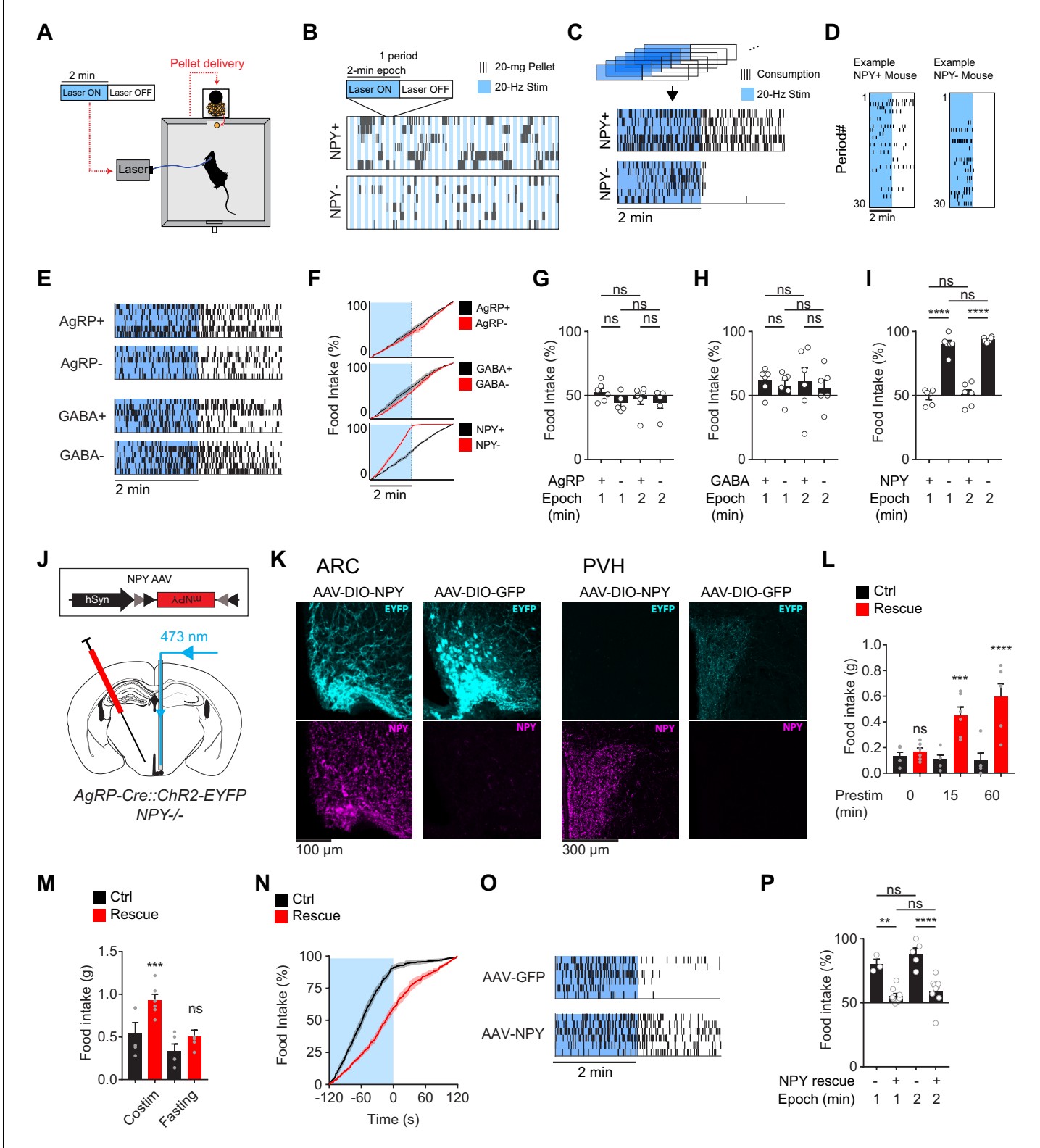

**Figure 3.** The requirement for NPY is rescued by reexpression of NPY in AgRP neurons. (a) Experimental setup for optogenetic stimulation with alternating laser ON and OFF epochs. (b) Pattern of pellet consumption for NPY +and NPY- mice in response to alternating laser ON and OFF epochs (2 min each). Each pair of laser ON and OFF epochs (period) is 4 min and the entire trial is 2 hr. Each row is a representative trial from an individual mouse. (c) Superimposed raster plots showing total pellet consumption of NPY+ and NPY- mice in relationship to the onset and offset of opto-

*Figure 3 continued on next page*

*Figure 3 continued*

stimulation for a 2 hr trial. Each row is a representative trial from an individual mouse. Raster plots were generated by superimposing the pellet consumption pattern from all periods of a trial. (**d**) Example of the temporal pattern of pellet consumption for a single NPY+ and NPY- mouse with successive periods of a single trial arranged vertically. (**e**) Superimposed raster plots of total pellet consumption across a 2 hr trial by AgRP+, AgRP-, GABA+, and GABA- mice in relationship to opto-stimulation pattern. (**f**) Cumulative pellet consumption by AgRP+ (N = 6), AgRP- (N = 5), GABA+ (N = 6), GABA- (N = 6), NPY+ (N = 5) and NPY- (N = 6) mice in relationship to opto-stimulation pattern. g-i, Percentage of food intake that occurs during the laser ON epochs of an intermittent stimulation trial by AgRP+ (N = 6) and AgRP- (N = 5) (**g**), GABA+ (N = 6) and GABA- (N = 6) (**h**), and NPY + (N = 5) and NPY- (N = 6) (**i**) mice. (**j**) Schematic for the rescue of NPY expression selectively in AgRP neurons. (**k**) Expression of EYFP (cyan) and NPY (magenta) in ARC (left) and PVH (right) of NPY- animals with or without NPY re-expression. Note that ChR2-EYFP is dim but visible in PVH of AAV-NPY injected animals. (**l**) Food intake (1 hr) by NPY rescue (red; N = 6) and control (black; N = 5) mice in response to different pre-stimulation protocols. (**m**) Food intake (1 hr) by NPY rescue (red; N = 6) and control (black; N = 5) groups in response to co-stimulation and overnight fasting (**n**) Cumulative pellet consumption by NPY rescue (N = 9) and control (N = 6) mice in relation to laser ON and OFF epochs. (**o**) Superimposed raster plots showing pellet consumption by NPY rescue and control mice in relation to the laser ON and OFF periods of a 2 hr trial. (**p**) Percentage of food intake by NPY rescue (N = 9) and control (N = 6) mice that occurs during the laser ON period. Filled area in f and n, error bars in g-i, l-m, and p represent mean ± SEM. Holm-Sidak multiple comparisons test was used to report adjusted P-values in g-i, l-m, and p. *p<0.05, **p<0.01, ***p<0.001, ****p<0.0001 or ns (not significant) compares adjacent control (black) and experimental (red) groups with same treatment, or compares pairs of groups indicated by the horizontal lines above.

DOI: https://doi.org/10.7554/eLife.46348.006

We then repeated this experiment in each of the genotypes described above. Remarkably, loss of NPY caused food intake to become strictly time-locked to the laser stimulus: that is, NPY- animals ate during concurrent stimulation, but then stopped eating within seconds of the laser shutting off (*Figure 3C,D,F,I*). In contrast, mice lacking GABA or AgRP signaling, as well as their littermate controls, showed no difference in feeding between the Laser ON and OFF periods (*Figure 3E–H*). These results were robust to variations in the paradigm, such as changing the duration of the stimulation epoch (*Figure 3G–I*). This indicates that, in the absence of NPY, AgRP neuron driven hunger dissipates within seconds of the offset of AgRP neuron firing.

NPY is expressed in cell types other than AgRP neurons. This raises the possibility that global NPY knockout could cause changes in other circuits that indirectly lead to deficits in AgRP neuron driven feeding. To exclude this possibility, we used a viral approach to rescue NPY expression selectively in AgRP neurons of NPY- mice. We generated two Cre-dependent AAVs, one expressing NPY and the other expressing GFP as a control, and then injected these AAVs into the ARC of NPY- mice (*Figure 3J*). Mice injected with AAV expressing NPY, but not GFP, showed restoration of NPY expression in the ARC (*Figure 3K*) and complete rescue of the feeding phenotypes associated with NPY- knockout (*Figure 3L–P*). For example, NPY re-expression in AgRP neurons caused feeding behavior in NPY- mice to become uncoupled from the onset and offset of laser stimulation, in a way that was indistinguishable from wild-type animals (*Figure 3N–P*). This indicates that the phenotypes observed in NPY- mice are due to loss of NPY expression in AgRP neurons, rather than effects of this mutation on other circuits or development.

We have focused here on the behavioral effects of AgRP neuron pre-stimulation (*Chen et al., 2016*). While this paradigm cannot replicate the precise details of asynchronous natural activity, it does mimic the broad features of the natural regulation of these cells (i.e. that AgRP neurons are more active before food discovery than during food consumption). In contrast, many earlier studies used a protocol in which AgRP neurons are stimulated concurrently with food consumption (*Aponte et al., 2011*; *Atasoy et al., 2012*; *Betley et al., 2013*; *Betley et al., 2015*). This protocol induces an activity pattern that is the opposite of what occurs naturally (i.e. greater AgRP neuron activity during feeding than before). To place our results in the context of this earlier work, we concurrently stimulated AgRP neurons using either chemogenetics (*Figure 4A–C*) or optogenetics (*Figure 4E–H*) and then measured food intake in NPY+ and NPY- mice. We found that concurrent chemogenetic stimulation increased food consumption in mice lacking NPY (*Figure 4C,D* and *Figure 4—figure supplement 1*), consistent with a previous report (*Krashes et al., 2013*). In contrast, during concurrent optogenetic stimulation NPY- mice consumed only a small fraction of the food consumed by NPY+ controls (*Figure 4E,F*). We could partially rescue food intake in NPY- mice by using a higher intensity optogenetic stimulation protocol (*Figure 4F*), whereas NPY+ controls showed no difference in food consumption between high and moderate intensity stimulation (high intensity refers to continuous 20 Hz stimulation, whereas moderate intensity refers to 20 Hz

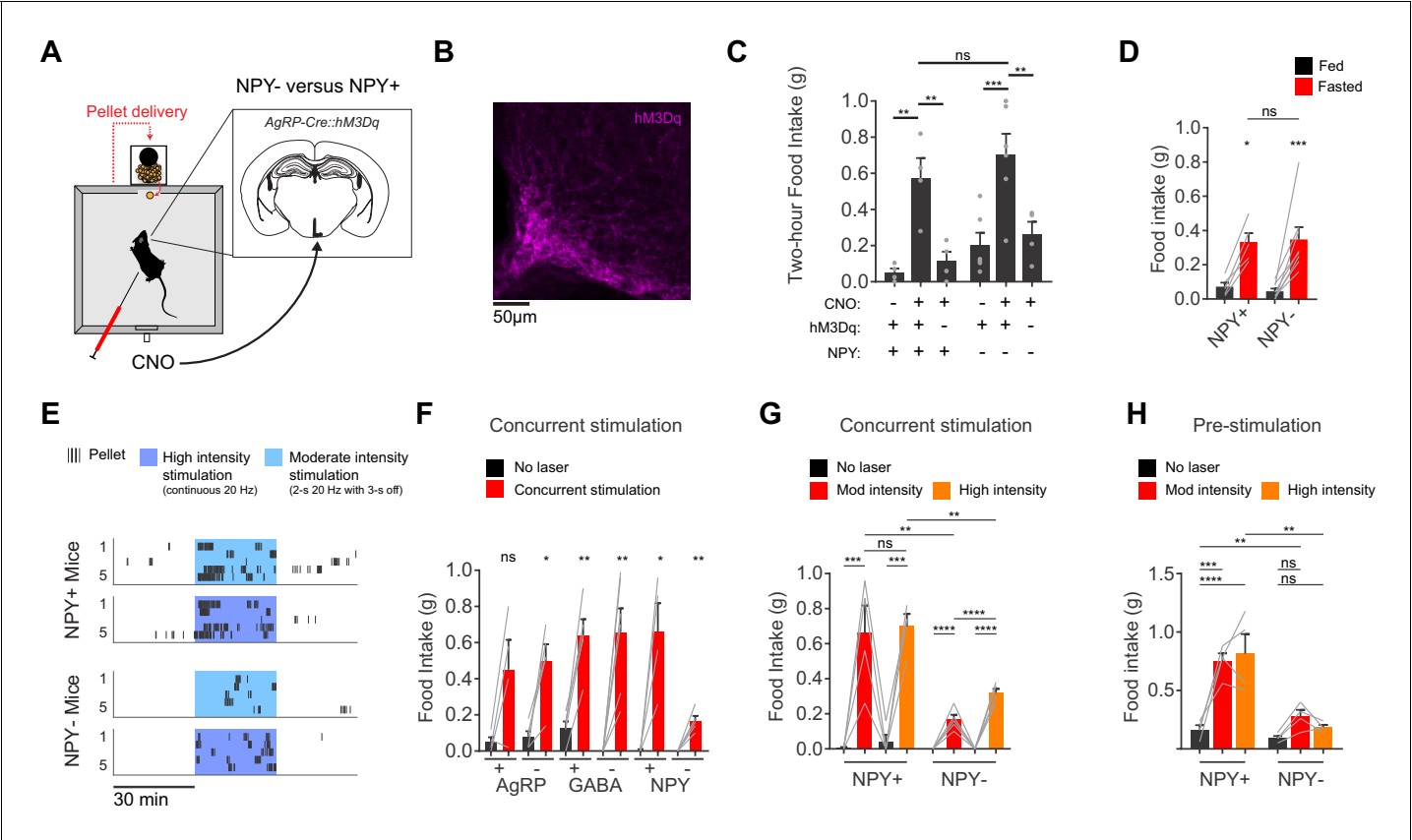

**Figure 4.** AgRP neurons show variable responses to concurrent stimulation in the absence of NPY. (a) Schematic for measurement of food intake during chemogenetic stimulation of AgRP neurons. (b) Expression of hM3Dq in AgRP neurons. Sample was stained for the HA tag on hM3Dq (magenta). (c) Food intake (2 hr) for NPY+ (hM3Dq+ N = 4; hM3Dq- N = 4) and NPY- (hM3Dq+ N = 6; hM3Dq- N = 4) mice following chemogenetic stimulation (CNO, 0.3 mg/kg) or control treatment. (d) Food intake (1 hr) for NPY+ (N = 5) and NPY- (N = 8) mice after overnight fasting. (e) Raster plot showing temporal pattern of pellet (20 mg) consumption by NPY- and NPY +mice in response to concurrent stimulation at high intensity (20 Hz, continuous) or moderate intensity (20 Hz, interleaved with 2 s ON/3 s OFF). (f) Food intake (30 min) for AgRP+ (N = 4) and AgRP- (N = 5), GABA+ (N = 5) and GABA- (N = 6), and NPY+ (N = 4) and NPY- (N = 5) mice in response to no stimulation (black) and concurrent stimulation (red; moderate-intensity). (g) Food intake (30 min) for NPY+ (N = 4) and NPY- (N = 5) mice in response to no stimulation (black), moderate intensity concurrent stimulation (red) and high intensity concurrent stimulation (orange). (h) Food intake (60 min) for NPY+ (N = 4) and NPY- (N = 4) mice in response to no stimulation (black), moderate intensity pre-stimulation (red) and high intensity pre-stimulation (orange). Error bars in c-e and g-j represent mean ± SEM. Holm-Sidak multiple comparisons test was used to report adjusted P-values in c-d and f-h. *p<0.05, **p<0.01, ***p<0.001, ****p<0.0001 or ns (not significant). Black lines denote the pairs that are being compared. In f, adjacent control (black) and experimental (red) groups subject to the same pre-stimulation protocol are compared.

DOI: https://doi.org/10.7554/eLife.46348.007

The following figure supplement is available for figure 4:

**Figure supplement 1.** Chemogenetic activation of AgRP neurons induces feeding in mice with or without NPY.

DOI: https://doi.org/10.7554/eLife.46348.008

stimulation in a 2 s ON, 3 s OFF pattern; *Figure 4G*). This indicates that intense concurrent optogenetic stimulation of AgRP neurons can partially overcome the requirement for NPY, likely through increased release of GABA (*Atasoy et al., 2012*; *Krashes et al., 2013*; *Tong et al., 2008*). It remains unclear why concurrent chemogenetic stimulation of AgRP neurons does not also require NPY to induce feeding (*Figure 4C,D* and *Krashes et al., 2013*). In contrast to these variable responses to concurrent stimulation, we found that increased stimulation intensity could not compensate to any degree for loss of NPY in a pre-stimulation paradigm (*Figure 4H*). This is presumably because endogenous GABA and AgRP are unable to drive food intake effectively on the timescale of sustained hunger (tens of minutes immediately after the offset of stimulation).

## Discussion

NPY was implicated in the regulation of feeding decades ago, but its precise contribution has been difficult to establish (*Loh et al., 2015*). Early studies showed that infusion of picomoles of NPY into the PVH can elicit voracious food intake (*Clark et al., 1985*; *Stanley and Leibowitz, 1984*), identifying NPY as an unusually potent orexigen. Consistent with this, NPY is highly expressed in hunger-promoting AgRP neurons (*Hahn et al., 1998*), and its expression in these cells is increased by energy deficit (*Sahu et al., 1988*; *White and Kershaw, 1990*). However genetic deletion of NPY, alone (*Erickson et al., 1996*) or in combination with AgRP (*Qian et al., 2002*), had little effect on food intake or body weight. Similarly deletion of NPY receptors, alone (*Kushi et al., 1998*; *Marsh et al., 1998*; *Pedrazzini et al., 1998*) or in combination (*Nguyen et al., 2012*), failed to produce the expected lean phenotype. More recently, the advent of cell-type-specific tools for neural manipulation made it possible to acutely stimulate AgRP neurons and then measure the effect of blocking NPY signaling. One study found that food intake following chemogenetic stimulation of AgRP neurons requires either NPY or GABA signaling, but not both (*Krashes et al., 2013*). Another study showed that antagonists of NPY or GABA receptors in PVH could partially block feeding induced by optogenetic stimulation of AgRP neurons (*Atasoy et al., 2012*). Taken together, these observations have led to a general view that NPY and GABA have partially redundant functions in mediating food intake by AgRP neurons.

The discovery that AgRP neurons are rapidly silenced by the sensory detection of food (*Betley et al., 2015*; *Chen et al., 2015*; *Mandelblat-Cerf et al., 2015*), and consequently can drive feeding via a long-lasting hunger signal (*Chen et al., 2016*), led us to wonder whether a specific neurotransmitter might mediate this unusual mechanism for controlling behavior. We therefore reinvestigated the role of these neurotransmitters using an optogenetic pre-stimulation paradigm that mimics the broad features of the natural regulation of AgRP neurons. This revealed a remarkably specific role for NPY in extending the duration of AgRP neurons' behavioral effects for tens of minutes beyond the window of their immediate firing. This in turn provides a molecular correlate for the long-lasting hunger signal that bridges AgRP neuron dynamics with feeding behavior.

While our findings have emphasized the role of NPY in sustained hunger, previous work has shown that the neuropeptide AgRP can also drive food intake over long timescales. For example, infusion of AgRP into the brain (*Hagan et al., 2000*) or stimulation of AgRP release (*Krashes et al., 2013*; *Nakajima et al., 2016*) can cause modulation of feeding behavior that persists for days. However, this long-lasting potentiation of feeding behavior requires several hours to emerge when endogenous AgRP release is stimulated (*Krashes et al., 2013*). Consistent with this, we find no evidence that the AgRP peptide is required for the intense food consumption that occurs immediately following AgRP neuron pre-stimulation (*Figure 1*). Thus our data suggest that NPY and AgRP play complementary roles in the regulation of feeding, with NPY sustaining hunger selectively on the timescale of a meal (tens of minutes after food discovery), whereas AgRP modulates the tone of the feeding circuit over longer timescales. This role for AgRP is consistent with an extensive literature that has implicated the melanocortin system in the long-term regulation of energy balance (*Krashes et al., 2016*).

A number of questions remain unanswered about how NPY mediates the sustained hunger response described here. Slice recordings have shown that NPY can have persistent effects on membrane excitability and neurotransmitter release in vitro (*Dubois et al., 2012*; *Fu et al., 2004*; *Roseberry et al., 2004*), indicating that NPY may act directly on post-synaptic targets of AgRP neurons to cause durable changes in their activity. We have shown that pre-stimulation of AgRP neuron axon terminals in PVH, BNST, and LH is sufficient to drive sustained feeding (*Chen et al., 2016*), suggesting that these are the downstream sites where NPY has its long-lasting effects on hunger. However it remains unknown which NPY receptors are involved, in which cell types these receptors are expressed, which molecular mechanisms they utilize, and how this sustained modulation is represented in the dynamics of the next circuit node. Likewise, while we have focused on NPY's role in appetitive and consummatory aspects of feeding, it remains unknown whether NPY has similar long-lasting effects on metabolism or other adaptive responses to energy state (*Dietrich et al., 2017*; *Padilla et al., 2017*; *Padilla et al., 2016*; *Steculorum et al., 2016*).

A central challenge of neuroscience is to explain how the neurochemical organization of the brain gives rise to behavior (*Bargmann and Marder, 2013*). Neural circuits contain an extraordinary

diversity of neuropeptides, yet in most cases we have little understanding of how those peptides relate to the function of the underlying cells. We have shown here how a neuropeptide that drives feeding can function on a timescale that is inaccessible to classical neurotransmitters. This in turn allows a neuron that controls hunger to direct ongoing behavior while simultaneously anticipating and responding to impending physiologic changes. It will be important to investigate to what extent other neuropeptides influence behavior in this way.

# Materials and methods

## Key resources table

| Reagent type (species) or resource | Designation | Source or reference | Identifiers | Additional information |
|---|---|---|---|---|
| Genetic reagent (Mus musculus) | B6; Agrp$^{tm1(cre)Lowl}$/J | Jackson Labs Stock | #012899 | |
| Genetic reagent (Mus musculus) | B6; 129S-Gt (ROSA)26Sor$^{tm32 (CAG-COP4*H134R/EYFP)Hze}$/J; | Jackson Labs Stock | #012569 | |
| Genetic reagent (Mus musculus) | B6; Tg (CAG-CHRM3, -mCitrine)1Ute/J | Jackson Labs Stock | #026220 | |
| Genetic reagent (Mus musculus) | B6; Cg-A$^y$/J | Jackson Labs Stock | #000021 | |
| Genetic reagent (Mus musculus) | Slc32a1$^{tm1Lowl}$/J | Jackson Labs Stock | #012897 | |
| Genetic reagent (Mus musculus) | 129S-Npy$^{tm1Rpa}$/J | Jackson Labs Stock | #004545 | |
| Antibody | anti-NPY (rabbit monoclonal) | Cell Signaling | 11976 | (1:1000) dilution |
| Recombinant DNA reagent | pAAV-EF1a-DIO-NPY | this paper | NA | plasmid with AAV backbone to express NPY in the presence of Cre recombinase |

## Mice

Mice were group housed on a 12:12 light:dark cycle with ad libitum access to water and mouse chow (PicoLab Rodent Diet 20, 5053 tablet, TestDiet). Adult mice (8–20 weeks old) were used for experiments. For channelrhodopsin-2 expression in AgRP neurons, Agrp-IRES-Cre mice (Jackson Labs stock 012899, B6; Agrp$^{tm1(cre)Lowl}$/J; RRID:IMSR_JAX:012899) were crossed with Ai32: ROSA26-loxStoplox-ChR2-eYFP mice (Jackson Labs stock 012569, B6; 129S-Gt(ROSA)26Sor$^{tm32(CAG-COP4*H134R/EYFP)Hze}$/J; RRID:IMSR_JAX:012569) to generate double mutant animals. For hM3Dq expression in AgRP neurons, Agrp-IRES-Cre mice were crossed with CAG-LSL-HA-hM3Dq-pta-mCitrine transgenic mice (Jackson Labs Stock 026220, B6; Tg(CAG-CHRM3,-mCitrine)1Ute/J.; RRID:IMSR_JAX: 026220) to generate double mutant animals. To block melanocortin signaling, mice were bred into heterozygotic A$^y$ background (Jackson Labs stock 000021, B6.Cg-A$^y$/J; RRID:IMSR_JAX: 000021); wild type cohorts were used as controls. To ablate GABA signaling specifically in AgRP neurons, mice were crossed into Agrp-IRES-Cre;Vgat$^{flox/flox}$ (Jackson Labs stock 012897, Slc32a1$^{tm1-Lowl}$/J; RRID:IMSR_JAX:012897) background; Agrp-IRES-Cre;Vgat$^{flox/+}$ animals were used as controls. To block NPY signaling, mice were crossed into NPY$^{-/-}$ background (Jackson Labs stock 004545, 129S-Npy$^{tm1Rpa}$/J; RRID:IMSR_JAX:004545); NPY$^{-/+}$ cohorts were used as controls.

Experimental protocols were approved by the University of California, San Francisco IACUC (Protocol AN133011) following the NIH guidelines for the Care and Use of Laboratory Animals.

## Stereotaxic viral delivery and fiber implantation

We performed intracranial surgery as previously described (Chen et al., 2015). For re-expression of NPY, pAAV-EF1a-DIO-NPY were generated from pAAV-EF1a-DIO-EYFP. Both vectors were made

into AAV8 in Stanford Vector Core. 200 nL of AAV8-EF1a-DIO-NPY or AAV8-EF1a-DIO-EYFP was bilaterally injected into the ARC (bregma: AP: −1.75 mm, ML: ±0.3 mm, DV: −5.9 mm). For optogenetic implants, custom-made fiberoptic implants (0.39 NA Ø200 μm core Thorlabs FT200UMT and CFLC230-10) were installed above the ARC (bregma: AP: −1.75 mm, DV: dorsal surface −5.6 mm, ML: −0.25 mm).

## Optogenetic food intake assay

To generate light pulses, a 473 nm laser was modulated by Coulbourn Graphic State software through a TTL signal generator (Coulbourn H03-14) and synchronized with behavioral experiments for each subject. The laser was passed through a single patch cable (Doric Lenses) to a custom-made fiber optic patch cable (Thorlabs FT200UMT, CFLC230-10; Fiber Instrument Sales F12774) through a rotary joint (Doric Lens FRJ 1 × 1). Patch cables were connected to the implant on each mouse through a zirconia mating sleeve (Thorlabs ADAL1). Laser power was set to 12–18 mW at the terminal of each patch cable unless otherwise specified.

Mice were allowed to recover for seven days after implant surgery before experiments. In addition to regular chow, mice were supplied *ad libitum* with the food pellets used during testing (20 mg Bio-Serv F0163) in their home cage. Mice were habituated to the behavioral chambers (Coulbourn H10-11M-TC with H10-11M-TC-NSF) and pellet dispensing systems (Coulbourn H14-01M-SP04 and H14-23M) for three days prior to the first experiment. Mice were provided *ad libitum* access to food and water unless otherwise specified and were tested during the early phase of the light cycle. All pre-stimulation food intake experiments followed this general structure: 70 min habituation/pre-stim period with no food access followed by 60 min of food access. Pellet removal was detected using a built-in photo-sensor (Coulbourn H20-93).

The following protocols were performed back to back to test the dose dependence of pre-stimulation on food intake. The sequence of testing was counter-balanced based on pre-stimulation duration.

| | | |
|---|---|---|
| 70 min habituation | 0 min opto-stim | 60 min food access |
| 65 min habituation | 5 min opto-stim | 60 min food access |
| 55 min habituation | 15 min opto-stim | 60 min food access |
| 40 min habituation | 30 min opto-stim | 60 min food access |
| 10 min habituation | 60 min opto-stim | 60 min food access |

[*]Laser was modulated at 20 Hz on a 2 s ON and 3 s OFF cycle with 10 ms pulse width.

The following protocols were performed back to back to test the decay kinetics of pre-stimulation on food intake. The sequence of testing was counter-balanced based on delay duration.

| | | | |
|---|---|---|---|
| 70 min habituation | 0 min opto-stim | 0 min delay | 60 min food access |
| 65 min habituation | 5 min opto-stim | 0 min delay | 60 min food access |
| 50 min habituation | 5 min opto-stim | 15 min delay | 60 min food access |
| 35 min habituation | 5 min opto-stim | 30 min delay | 60 min food access |
| 5 min habituation | 5 min opto-stim | 60 min delay | 60 min food access |

[*]Laser was modulated at 20 Hz on a 2 s ON and 3 s OFF cycle with 10 ms pulse width.

The following protocols were performed back to back to test the kinetics of the sustained hunger signal on the timescale of tens of seconds.

| | |
|---|---|
| 2 min opto-stim | 2 min no-stim |

Repeated 30 times. Food always available.

| | |
|---|---|
| 1 min opto-stim | 1 min no-stim |

Repeated 60 times. Food always available.

[*]Laser was modulated at 20 Hz with 10 ms pulse width.
The following protocols were performed back to back to test the effect of concurrent stimulation and its interaction with stimulation intensity.

| 30 min no-stim | 30 min opto-stim | 30 min no-stim |
| --- | --- | --- |

Food always available. Laser was modulated at 20 Hz with 10 ms pulse width.

| 30 min no-stim | 30 min opto-stim | 30 min no-stim |
| --- | --- | --- |

Food always available. Laser was modulated at 20 Hz on a 2 s ON and 3 s OFF cycle with 10 ms pulse width.

## DREADDs food intake assay

Housing and habituation of animals was identical to the optogenetic food intake assays. Mice were additionally habituated to handling for I.P injection twice before actual experiments. After habituation, mice were tested at most once every day during light phase for food intake (2 hr). Mice were injected with CNO (0.3 mg/kg) or vehicle (0.01 mg/kg DMSO) at the beginning of each food intake assay (2 hr).

## Progressive ratio testing

For training, mice were acutely food deprived 5 hr before the start of dark cycle and trained with fixed-ratio 1 (FR1), FR3, and FR7 protocols overnight until active lever presses exceeded learning thresholds, which were 50, 150, and 300 respectively. Of note, NPY- mice were trained for longer periods due to their slower learning rate that was partially caused by their lack of exploration in the operant chamber. Mice were then acutely food deprived overnight and tested with a progressive ratio 3 (PR3) task for 1.5 hr at the start of the dark phase.

For optogenetic experiments, during the first 70 min of the testing protocol (habituation/pre-stim), access to the levers and pellet trough was blocked using a custom-cut acrylic board. After the initial 70 min, the acrylic board was removed, and a single pellet was delivered following active lever presses according to the PR3 schedule (3,6,9,12. . .). Each experiment was subject to multiple repetitions to test robustness.

## Immunohistochemistry

Immunofluorescence was performed as previously described (*Chen et al., 2015*) using the following pairs of primary and secondary antibodies: chicken anti-GFP (1:1000; Abcam Cat# ab13970, RRID: AB_300798) and polyclonal donkey anti-chicken FITC (1:1000; Abcam Cat# ab63507, RRID:AB_1139472); goat anti-Fos (1:500; Santa Cruz Biotechnology Cat# sc-52-G, RRID:AB_2629503) and donkey anti-goat Alexa-fluorophore 568 (1:1000; Thermo Fisher Scientific Cat# A-11057, RRID:AB_2534104); rabbit anti-NPY (1:1000; Cell Signaling Technology Cat# 11976, RRID:AB_2716286) and donkey anti-rabbit Alexa-fluorophore 647 (1:1000; Thermo Fisher Scientific Cat# A-31573, RRID:AB_2536183); goat anti-HA (1:1000; for validation of HA-tagged hM3Dq expression; Abcam Cat# ab9134, RRID:AB_307035) and donkey anti-goat Alexa-fluorophore 568 (1:1000; Thermo Fisher Scientific Cat# A-11057, RRID:AB_2534104).

## Statistics

Raw behavioral data were analyzed with custom MATLAB scripts to obtain time-stamped behavioral events. Multiple measurements from the same mouse in the same experiment (e.g. on different days) were considered technical repeats and were averaged before statistical analyses. Control experiments were repeated at the beginning and end of each set of assays. Experiments with technical failures (e.g. dysfunction of the behavioral system, pellets stuck in feeder, broken fiberoptic patch cables) were excluded from further analysis and repeated. The average of these technical repeats for each mouse in each experiment was considered a single biological repeat. All sample sizes are numbers of biological repeats. Data was analyzed by two-way ANOVA using Graphpad Prism seven to test for an effect of genotype and stimulation protocol or one-way ANOVA. Individual p-values were

corrected using Holm-Sidak's multiple comparisons test. Regression analysis for experiments investigating feeding kinetics was performed using Graphpad.

## Acknowledgements

YC is supported by a Howard Hughes Medical Institute International Student Fellowship. ZAK is a Howard Hughes Medical Institute Investigator and this work was supported by the New York Stem Cell Foundation, American Diabetes Association Pathway Program, Rita Allen Foundation, McKnight Foundation, Alfred P Sloan Foundation, Brain and Behavior Research Foundation, Esther A and Joseph Klingenstein Foundation, UCSF Program for Breakthrough Biomedical Research, UCSF Diabetes Center, UCSF Nutrition Obesity Research Center, and NIH grants DP2-DK109533, R01DK106399 and R01NS094781.

## Additional information

### Funding

| Funder | Grant reference number | Author |
|---|---|---|
| Howard Hughes Medical Institute | International Student Fellowship | Yiming Chen |
| National Institute of Diabetes & Digestive & Kidney Diseases | 1F32DK118843 | Oliver H Miller |
| National Institutes of Health | R01DK106399 | Zachary A Knight |
| National Institutes of Health | R01NS094781 | Zachary A Knight |
| Howard Hughes Medical Institute | | Zachary A Knight |
| American Diabetes Association | ADA Accelerator Grant | Zachary A Knight |
| Rita Allen Foundation | | Zachary A Knight |
| New York Stem Cell Foundation | | Zachary A Knight |
| National Institutes of Health | 1DP2DK109533 | Zachary A Knight |
| National Institutes of Health | 5P30DK098722 | Zachary A Knight |

The funders had no role in study design, data collection and interpretation, or the decision to submit the work for publication.

### Author contributions

Yiming Chen, Conceptualization, Data curation, Funding acquisition, Investigation, Methodology, Writing—original draft, Writing—review and editing; Rachel A Essner, Data curation, Investigation, Methodology; Seher Kosar, Oliver H Miller, Yen-Chu Lin, Sheyda Mesgarzadeh, Investigation; Zachary A Knight, Conceptualization, Resources, Supervision, Funding acquisition, Writing—original draft, Project administration, Writing—review and editing

### Author ORCIDs

Yiming Chen [iD] https://orcid.org/0000-0002-9569-1929
Rachel A Essner [iD] https://orcid.org/0000-0003-2240-3911
Zachary A Knight [iD] http://orcid.org/0000-0001-7621-1478

### Decision letter and Author response

Decision letter https://doi.org/10.7554/eLife.46348.012
Author response https://doi.org/10.7554/eLife.46348.013

## Additional files

### Supplementary files
• Transparent reporting form
DOI: https://doi.org/10.7554/eLife.46348.015

### Data availability
All data generated or analysed during this study are included in the manuscript and supporting files.

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
