## [Decision Letter]

Thank you for submitting your article "Sustained NPY signaling enables AgRP neurons to drive feeding" for consideration by *eLife*. Your article has been reviewed by two peer reviewers, including Richard D Palmiter as the Reviewing Editor and Reviewer #1, and the evaluation has been overseen by Catherine Dulac as the Senior Editor. The following individual involved in review of your submission has also agreed to reveal his identity: Michael J Krashes (Reviewer #2).

The reviewers have discussed the reviews with one another and the Reviewing Editor has drafted this decision to help you prepare a revised submission.

The two reviewers both conclude that this paper represents a significant increase in understanding of the underlying signaling molecules responsible for the persistent feeding response after optogenetic stimulation of AgRP neurons ceases. The authors convincingly show that only neuropeptide Y (rather than AgRP or GABA) can drive the feeding response. No further experiments are necessary, but there are some points that indicated below that should be addressed in a revised version.

*Reviewer #1:*

This paper represents a follow-up on the authors' previous paper (Chen et al., 2016) in which they discovered that there is a persistent effect on food intake when AgRP neurons in sated mice are stimulated (ChR2) without access to food and then food is made available without further stimulation. Here using mice in which either NPY, AgRP or GABA signaling is blocked) they demonstrate that only signaling by NPY from AgRP neurons (not AgRP or GABA) allows this persistent food intake without concomitant stimulation. They demonstrate this important point several different ways. The results are compelling and described clearly. This is a nice example of how a neuropeptide can have persistent activity even after its release is terminated. The Discussion is cogent and to the point.

*Reviewer #2:*

This manuscript by Chen et al., seeks to evaluate the role of NPY released from AgRP neurons in the regulation of feeding behavior. Although decades of research have convincingly demonstrated its orexigenic role, a new investigation is warranted given recent findings that pertain to the endogenous activity of AgRP neurons in response to caloric information. This manuscript is clearly designed to do just that, finding that NPY is uniquely required for the continuing effects of AgRP neurons on feeding behavior. Importantly these effects are specific for NPY and not GABA or AgRP. My comments for improving the manuscript are listed below:

Please visualize the data presented in bar graphs as individual data points as opposed to the SEM lines. This gives the reader a clearer indication of the variability and number of mice used without having to look this information up elsewhere.

Detail regarding the low versus high intensity photostimulation should be provided in the text. The methods indicate that low stimulation is already 12-18 mW (which in my estimation is already quite high). The authors state that they are able to partially rescue food intake in NPY- mice by using a higher intensity opto protocol by proposing the higher intensity stim leads to amplified GABA release. Has this phenomenon been shown in AgRP neurons, or any other group of cells? If so, it should be referred to here.

The authors show that there does not seem to be a defect in refeeding after a fast in NPY- mice compared to littermate hets (Figure 4D). However, this 1 hr food intake measurement seems low for this metric. Could the authors comment on this? Do the NPY KO hets have a defect in refeeding according to the original report or is this just the result of this specific experiment?

Perhaps the biggest surprise that deserves further discussion is the differences observed between chemo and opto experiments in Figure 4. Firstly, the authors should point out in the text that the chemogenetic food intake measurement was conducted over 2 hours versus the optogenetic experiments (30 mins or 1 hour). Thus the increase in food intake observed in the chemogenetic experiment could be merely due to the longer amount of time the animals had to eat. As written currently, it is unclear whether the approaches elicit distinct release properties (this may be the case but they should cite instances where differences in manipulation techniques elicited different responses) or whether these inconsistencies are due to temporal measurements. Either way, it is interesting and should be discussed further.

Some comment on the non-physiological synchronous ChR2-mediated activation of AgRP neurons compared to their endogenous activity should be highlighted with interpretation.

---

## [Author Response]

Reviewer #2:This manuscript by Chen et al., seeks to evaluate the role of NPY released from AgRP neurons in the regulation of feeding behavior. Although decades of research have convincingly demonstrated its orexigenic role, a new investigation is warranted given recent findings that pertain to the endogenous activity of AgRP neurons in response to caloric information. This manuscript is clearly designed to do just that, finding that NPY is uniquely required for the continuing effects of AgRP neurons on feeding behavior. Importantly these effects are specific for NPY and not GABA or AgRP. My comments for improving the manuscript are listed below:Please visualize the data presented in bar graphs as individual data points as opposed to the SEM lines. This gives the reader a clearer indication of the variability and number of mice used without having to look this information up elsewhere.

We have added individual data points/lines on top of bar graphs to indicate replicates.

Detail regarding the low versus high intensity photostimulation should be provided in the text. The methods indicate that low stimulation is already 12-18 mW (which in my estimation is already quite high).

The terms “low intensity” vs. “high intensity” in the text refer to the number of laser pulses, not the power level. High intensity is continual 20 Hz stimulation. Low intensity is 20 Hz in a 2s ON: 3s OFF pattern. We have clarified this in the text. We also agree with the reviewer that 12-18 mW is not a low laser power. We have changed the terminology in the revised text so that the 2s ON: 3s OFF stimulation is referred to as “moderate intensity” instead of “low intensity” (copied below).

“…whereas NPY+ controls showed no difference in food consumption between high and moderate intensity stimulation (high intensity refers to continuous 20 Hz stimulation, whereas moderate intensity refers to 20 Hz stimulation in a 2-s ON, 3-s OFF pattern; Figure 4G).”

The authors state that they are able to partially rescue food intake in NPY- mice by using a higher intensity opto protocol by proposing the higher intensity stim leads to amplified GABA release. Has this phenomenon been shown in AgRP neurons, or any other group of cells? If so, it should be referred to here.

We are simply suggesting that higher frequency (not higher power) stimulation will lead to a higher frequency of action potentials, which will then lead to an increase in neurotransmitter release. For AgRP neurons, this has been demonstrated in Atasoy et al., 2012, albeit with a somewhat different protocol.

The authors show that there does not seem to be a defect in refeeding after a fast in NPY- mice compared to littermate hets (Figure 4D). However, this 1 hr food intake measurement seems low for this metric. Could the authors comment on this? Do the NPY KO hets have a defect in refeeding according to the original report or is this just the result of this specific experiment?

We also noticed that the amount of food intake consumed was low. However this value is not too different from what was reported in another study using NPY KO animals (e.g. Figure 1 in Bannon et al. Brain Research 2000, reference below). Moreover, certain studies (Bannon et al. Brain Research 2000) but not others (Erickson et al., Science 1996) have observed differences between NPY-/- and NPY+/? animals in fasting-induced refeeding. We think this variability likely reflects a number of factors, including strain background. For this reason we were careful to use littermates as controls in every experiment with a mutant animal.

Erickson, J. C., Clegg, K. E., and Palmiter, R. D. (1996). Sensitivity to leptin and susceptibility to seizures of mice lacking neuropeptide Y. Nature. https://doi.org/10.1038/381415a0

Bannon, A. W., Seda, J., Carmouche, M., Francis, J. M., Norman, M. H., Karbon, B., and McCaleb, M. L. (2000). Behavioral characterization of neuropeptide Y knockout mice. Brain Research, 868(1), 79–87. https://doi.org/10.1016/S0006-8993(00)02285-X

Perhaps the biggest surprise that deserves further discussion is the differences observed between chemo and opto experiments in Figure 4. Firstly, the authors should point out in the text that the chemogenetic food intake measurement was conducted over 2 hours versus the optogenetic experiments (30 mins or 1 hour). Thus the increase in food intake observed in the chemogenetic experiment could be merely due to the longer amount of time the animals had to eat.

We plotted 30 and 60 min timepoints for the optogenetic experiments because in the optogenetic experiments the mice eat most of the food in the first hour. We plotted two hours for the chemogenetic experiments because the mice eat slower. To allow direct comparison, we have included in the revised manuscript the entire timecourse of feeding for the chemogenetic experiments as well as a bar graph comparing the 60 min food intake data for the chemogenetic experiments (Figure 4—figure supplement 2). There is no significant difference between the genotypes in their response to CNO at any timepoint.

As written currently, it is unclear whether the approaches elicit distinct release properties (this may be the case but they should cite instances where differences in manipulation techniques elicited different responses) or whether these inconsistencies are due to temporal measurements. Either way, it is interesting and should be discussed further.

As stated above, the inconsistencies between the optogenetic and chemogenetic results are not due to the timing of the measurements. We don’t know the reason for this difference. We have inserted a sentence stating this clearly:

“It remains unclear why concurrent chemogenetic stimulation of AgRP neurons does not require NPY to induce feeding (Figure 4C, D and (Krashes, 2013)).”

Some comment on the non-physiological synchronous ChR2-mediated activation of AgRP neurons compared to their endogenous activity should be highlighted with interpretation.

We have added the additional clause below.

“We have focused here on the behavioral effects of AgRP neuron pre-stimulation (Chen et al., 2016). While this paradigm cannot replicate the precise details of asynchronous natural activity, it does mimic the broad features of the natural regulation of these cells (i.e. that AgRP neurons are more active before food discovery than during food consumption).”

We have avoided further speculation about the importance of asynchronous AgRP activation, in part because previous studies have suggested that optogenetic stimulation of AgRP neurons does not induce typical synchronous activity. For example see Figure 1D and pg. 20 in (Mandelblat-Cerf et al., 2015). ‘While many AgRP neurons showed classical entrainment to the pulse train at 20 Hz, some clearly laser driven AgRP neurons did not show strong entrainment.’